# Tap Water Consumption Is Associated with Schoolchildren’s Cognitive Deficits in Afghanistan

**DOI:** 10.3390/ijerph19148252

**Published:** 2022-07-06

**Authors:** Abdullah Shinwari, Alain Véron, Mohammad Haris Abdianwall, Elisabeth Jouve, Remi Laporte

**Affiliations:** 1Department of Public Health, Nangarhar Medical College, Nangarhar University, Jalalabad 2601, Afghanistan; drabdullah172@gmail.com (A.S.); haris_abdianwall@yahoo.com.tr (M.H.A.); 2CEREGE, Aix-Marseille University, CNRS, IRD, College de France, INRAe, Europôle Méditerranéen de l’Arbois, BP80, CEDEX 4, 13545 Aix en Provence, France; 3Service d’Evaluation Médicale, Assistance Publique—Hôpitaux Marseille, 13005 Marseille, France; elisabeth.jouve@ap-hm.fr; 4Equipe d’Accueil 3279-Centre d’Études et de Recherche sur les Services de Santé et la Qualité de Vie (CEReSS), Aix Marseille University, Laboratoire de Santé Publique, Faculté de Médecine, 27 Boulevard Jean Moulin, CEDEX 5, 13385 Marseille, France; remijulien.laporte@ap-hm.fr

**Keywords:** schoolchildren, TONI-1, intelligence quotient, age, tap water, Afghanistan

## Abstract

Environmental influence on intelligence quotient (IQ) is poorly understood in developing countries. We conducted the first cross-sectional investigation to assess the role of socio-economic and environmental factors on schoolchildren’s IQ in Jalalabad, Afghanistan. A representative sample of 245 schoolchildren aged 7–15 was randomly selected in five schools. Children’s records included: non-verbal IQ TONI-1 scale, body mass index, socio-economic status, and further environmental indicators (water supply, proximity to a heavy-traffic road, use of surma traditional cosmetics). The mean age of the children was 11.7 years old (±2.0 years), and 70.2% and 29.8% were male and female, respectively. The children’s mean IQ was 83.8 (±12.6). In total, 37 (14.9%) of the children were overweight, 78 (31.5%) were living below the USD 1.25 poverty line, 133 (53.6%) used tap water supply, 76 (30.6%) used surma, and 166 (66.9%) were exposed to heavy road traffic. The children’s IQ was significantly and independently lowered by tap water use (−3.9; 95% CI [−7.1; −0.6]) and by aging (−1.4; 95% CI [−2.2; −0.6]), as revealed in multivariate analysis, independently of gender, socio-economic status, exposure to heavy road traffic, overweight status, and surma use. Lower IQ among older children is possibly attributed to chronic stress experienced by adolescents due to living conditions in Afghanistan. While using tap water prevents fecal peril, it may expose children to toxic elements such as lead which is known to lower their intellectual development.

## 1. Introduction

In developing countries, 200 million children aged 5 or less are hindered from reaching their full cognitive development and school achievements [1] because of social health inequalities [2,3,4]. Their cognitive abilities are plagued by political, socio-economic, educational, and environmental issues that include poverty, malnutrition, violence exposure, and toxic element hazards among the most prominent casualties [5,6,7,8,9,10,11,12,13,14,15,16,17,18,19,20]. Malnutrition is a major health issue as it increases childhood morbidity and mortality (6 million deaths per year in developing countries) and leads to permanent impairment of physical and mental growth [21], most particularly in low-income countries (LICs) [2,3,22,23,24,25,26]. Socio-economic status (as expressed by poverty status) is the other well-known predictor to impeding children’s cognitive development, as evidenced in LICs such as Chile, Iran, Nepal, India, and Malaysia [2,3,27,28,29,30]. Environmental issues may also delay brain maturing owing to the ingestion of neurotoxic metals that are known to interfere with the central nervous system [31,32,33,34,35]. Cognitive development can be assessed with “Intelligence Quotient” (IQ), closely correlated to nutrition and economic status in LICs [22,23,24,25,27,28,29,36]. Stunted children are more likely to have poorer cognitive ability or achievement scores [30,37].

In Afghanistan, children face most of these threats because of devastating war, long-lasting instability, and administrative weakness. Afghanistan is among the least-developed countries according to the United Nations’ *Human Development Index* and the World Bank’s *Human Capital Index,* with 54.6% of the population living below the national poverty line during the 2008–2019 period [38,39,40]. Afghan children are afflicted by significant health issues: substandard sanitation (with unsafe drinking water and fecal peril), improper and hazardous solid waste management (with industrial and agricultural toxic exposures), macro- and micro-nutrient deficiency, and insufficient food hygiene practices [41]. In recent years, 600,000 Afghan children were suffering from severe acute malnutrition [21].

To our knowledge, there are no existing studies about the consequences of these living conditions on children’s cognitive development in Afghanistan. This study aimed to describe and distinguish the impact on children’s IQ of these adverse prominent predictors, i.e., nutritional status, poverty status, and environmental exposures.

## 2. Material and Method

### 2.1. Children’s Cohort

This cross-sectional multicentric study was conducted in November 2019 among schoolchildren in Jalalabad city, the capital of Nangarhar province in eastern Afghanistan (Figure 1). Jalalabad has 70 schools (20 public and 50 private). Children at school were chosen to participate in this study rather than children at home in order to allow examination of both male and female children from the full socio-economic spectrum of children attending school. The sampling method for the survey was a two-stage random sampling. The necessary representative number of children within the cohort was determined using the Epi-info software (with 95% confidence interval, with a total population of 98,272 schoolchildren). A sample of 245 children was randomly selected among 5 out of 70 schools, leading to 49 students from each school. The sample size was increased to 15% to take into account the risk of refusal to participate and missing data. School sampling was addressed with the Education Directorate Board of Nangarhar province because of feasibility limits. The five following schools were retained: Qandhara “S1”, Iqra “S2”, Afghan International “S3”, Bibi Zainab “S4”, and Al-Taqwa “S5” (Figure 1). Four schools were private institutes while school 3 (S3) was a public female school. All children aged 7–15 were included in the study apart from those who refused to participate and those who suffered from genetic, congenital, or acquired chronic diseases. A total of 77% of the children were above 10 years old. During interviews, children were properly briefed by being given examples and diagrams for their understanding, so that they were able to answer the environmental exposure factors and other questions.

### 2.2. Cognitive Abilities

Children’s cognitive abilities were assessed with the TONI-1 non-verbal IQ test. This measure of general intelligence is a progressive matrices method designed to rate abstract/figural problem-solving skills of children independently of language and schooling [42,43,44,45]. The non-verbal TONI-1 test has been recognized as among the best recommended tests owing to its psychometric soundness for low-stakes assessments in the general population [46]. The TONI-1 test was administered to the children in a quiet, cozy, well-lit room. They were acclimatized to the test with a 10-minute demonstration through six training items before performing the test during a period of 30 min. IQ raw scores were classified into Superior (131–145), Above Average (116–130), Average (85–115), Below Average (70–84), and Low/Borderline (55–69).

### 2.3. Main Predictors

Nutrition status was assessed using the body mass index (BMI). BMI was determined from the weight, height, and age of the children. The weight of the children was taken in kilograms by using a portable electronic digital weighing scale provided by a pediatric clinic in Jalalabad city. The height of the children was taken in meters with a portable direct-reading stadiometer. The survey team consisted of a doctor and three well-trained nurses. BMI was calculated from weight divided by height, converted into age z-score using WHO charts for 5–19-year-old children, according to overweight and obesity (>+2 SD) and thinness (<−2 SD) [47].

Poverty status was assessed using the people poverty index (PPI) scale developed by the Grameen Foundation. This individual-level poverty index comprehends a set of 10 country-specific indicators covering household characteristics. PPI does not require direct questions, but rather an indirect assessment of socio-economic status by asking easy and simple questions such as how many children go to school and which household assets are owned including a car (Appendix A). PPI allows estimation of the likelihood of a household living below international or national poverty lines, according to the World Bank’s international poverty lines such as the purchasing power parity of USD 1.25/day, or the USAID extreme poverty line and the national poverty line for each country. In this study, the international poverty line of USD 1.25/day was used because the World Bank decided that people living below USD 1.25/day cannot afford the basic needs a person requires to live and therefore is considered in extreme poverty. Experts evaluated what could be purchased in the US market for USD 1.25 and afterward determined how much that equivalent “basket of goods” would cost in various other countries. Therefore, less than USD 1.25/day is considered a standard measure of poverty and is an international standard that can be used in various countries and compared to other organization’s standards.

Exposure to toxic elements was appraised by several variables: a heavy-traffic road within 100 m of the home for atmospheric pollution, use of potentially traditional lead-contaminated cosmetics (surma), and type of water supply (tap water versus local pumping devices). These exposures may be responsible for trace metal ingestion or inhalation that is known to impair children’s cognitive development [48,49,50]. Water supply was categorized into two categories reflecting two risk–benefit ratios: (i) tap water can offer protection against fecal peril but is potentially exposed to water contamination by toxic trace metals during storage and shipping in pipelines and tanks [51,52,53,54,55]; and (ii) local water drawing (well, spring, hand pump, or river) is potentially exposed to fecal peril infectious diseases but can offer protection against toxic metal exposure [56].

### 2.4. Statistical Analysis

Descriptive statistics were calculated for each dependent variable by mean and standard deviation (±SD), or frequency. IQ scores are presented as a box-whisker plot showing a plot for each school and one for all schools. The chi-square test or Student t-test was used for comparison of predictors between schools. Pairwise comparisons between schools were performed using Tukey’s test for means or the Bonferroni adjustment for percentages. A mixed model was used to examine the association between the IQ score and predictors. The school variable was included as a random effect, accounting for the additional component of variance associated with a cluster sampling design to overdraw any assumption of data correlation between children in the same school. Significance was set at *p*-value < 0.05. Analyses were performed using IBM SPSS Statistics Software, version 21 (IBM Corp., Armonk, NY, USA) [57].

## 3. Results

All the children’s data are presented in Appendix A for each school with detailed information for each child referenced as a number in column A of the Excel table. During interviews with the children, data on the “water source” predictor was initially displayed as five categories to fit with individual descriptions: ducted from long distance and stored for tap water, nearby distributed well, hand pump, spring water, and riverine water. Meanwhile, local findings showed classification biases in responses (excluded nearby distributed well, hand pump, spring, and riverine water). Thus, for analysis of water distribution and storage mode, we subsequently classified water usage as tap water usage versus any other local water collection. The children’s mother language is also presented, mostly comprising two main mother tongues—Pashtoon and Tajik—showing the relative cultural uniformity of the cohort.

### 3.1. Descriptive Analysis

The mean age of the children was 11.7 years old (±2.0 years), and 70.2% and 29.8% were male and female, respectively. Children aged 10 or higher comprised 77.0% of all children. A total of 28 children were non-respondents because of parental refusal, causing additional sampling and an uneven number of children in each school cohort (Table 1). The rate of refusal was different among schools (*p* < 0.0001), the highest being in S2 (28.1%) and S5 (21.0%). Environmental chemical exposure identified 166 (66.9%) children living close to a heavy-traffic road, 76 (30.6%) endorsing the use of surma cosmetics, and 133 (53.6%) the use of tap water. There were significant differences between schools for age, gender, use of surma, and tap water predictors (Table 1). The age in school 2 was significantly lower than in school 3 (*p* < 0.005) and 4 (*p* < 0.05), and the BMI z-score in school 2 was significantly higher than in school 3 (*p* < 0.05). The use of surma was higher in school 4 than in school 2 (*p* < 0.05). The use of tap water was higher in schools 1 and 2 than in schools 4 and 5 (*p* < 0.01), and higher in school 5 than in school 3 (*p* < 0.01). The mean IQ was significantly lower in school 3 than in schools 4 and 5 (*p* < 0.05).

Children’s IQ score are presented in Figure 2 with 122 (49.2%) children being classified as “Average” IQ, 86 (34.6%) and 40 (16.3%) as “Below Average” and “Low/Borderline” IQ, respectively. No child was “Above Average” and 50.9% were under the “Below Average” line when all schools were considered.

Medians of the BMI z-score were within the normal range for 7–9 and 10–15 age groups, the 7–9 group being significantly lower than the 10–15 group. Overall, 12.7% of the children were overweight with only one being classified as thin. The mean height-for-age z-score was −0.46 (±1.34). Only 22 (8.9%) children had a short stature (<−2 standard deviation). It was neither different by school (*p* = 0.37) nor correlated with IQ (Pearson r^2^ = 0.04; *p* = 0.59). However, it was correlated with age (Pearson r^2^ = 0.43; *p* < 0.0005).

### 3.2. Statistical Analysis

In a bivariate analysis, a lower IQ score was significantly associated with older age and poverty status (*p* < 0.05). There was a trend, although non-significant, of lower IQ score association with tap water use (*p* = 0.06) (Table 2). Other predictors showed no significant association with IQ.

Multivariate mixed-model analysis showed that tap water use and older age were significantly and independently associated with lower IQ (Table 2). This model allowed an adjustment with other covariates: gender, poverty status, overweight, heavy road traffic, and surma use.

## 4. Discussion

Our original results showed a high frequency of low IQ among schoolchildren in Afghanistan. Lower IQ was significantly and independently associated with tap water use and with higher age.

Our mean TONI-1 IQ is in the lower range of 13 Asian countries when compared with national IQs reported by Lynn [58]. The methodology employed by Lynn et al. [59,60] and their national IQ estimates should be used with caution, particularly because IQ tests have been shown not to be culturally neutral in the sub-Saharan regions. Their IQs may have been misestimated (10 to 15 points) when compared to other methodologies [61,62]. Because of cultural imprints and the multiplicity of IQ tests, the comparison of our IQs to other regional or national IQs would require complex adjustments that go beyond the scope of this study [63]. On the contrary, the TONI-1 test we used did not show such cultural biases. The non-verbal TONI-1 test has been recognized among the best recommended tests owing to its psychometric soundness for low-stakes assessments in the general population [46]. Unfortunately, to our knowledge, large studies using the TONI-1 test are still rare for comparisons between Asian countries. Thus, our original IQ results showing a high frequency of low IQ among children in Afghanistan may best reflect a tragic but real regional situation.

Results show that schoolchildren’s IQ in Jalalabad is significantly and independently lowered by tap water use and age, independently from other predictors (gender, socio-economic status, exposure to heavy road traffic, overweight status, and surma use). The BMI z-score is known to be correlated with recent nutrition status, and height-for-age z-score with chronic malnutrition. The age variable was chosen to be included in the multivariate model to handle the whole allostatic load (including long-term nutritional status). Most particularly, nutrition deficiency was not an issue for our children cohort and seemed to have no negative effect on the children’s IQ.

Tap water infrastructures are widely designed to avoid waterborne diseases in developing countries [56,64,65]. The water supply in Afghanistan often originates from surface water, i.e., local sources and wells with potential exposure to nitrates, borates, and fecal bacteria that often exceed permissible limits. This is particularly true in cities, owing to uncontrolled sewage spreading, the use of fertilizers and unconfined landfills, and the overexploitation of groundwater [66,67]. Only wealthy urban households can afford water provided by the Afghan Urban Water Supply and Sewerage Corporation [68]. In the early 2000s, less than 20% of the population had access to piped water in Kabul [69]. While safer against fecal peril, pipes and end-point devices (brass plumbing) used for tap water supply may release significant amounts of trace metals such as lead, zinc and cadmium [53,70,71,72,73,74,75,76]. Tap water supply is often polluted with trace metals in Afghanistan’s capital city, Kabul, due to corroded pipes [77]. Furthermore, domestic water tanks are widely used for tap water because of intermittent water supply. Pollution of drinking water increases with low quality of the tank material and the length of storage [78,79,80,81,82,83,84]. As a result, drinking water may contain higher concentrations of several trace metals: iron, copper, zinc, lead, and manganese, above WHO standards, when stored for months in galvanized steel, polyethylene, or concrete tanks [54,84,85,86,87]. To our knowledge, no study has yet addressed this issue in Afghanistan. Neurotoxicological and cognitive effects are clearly demonstrated in children exposed to lead contamination [33,88,89,90,91]. Cognitive deficiencies have also been evidenced for manganese, cadmium, and arsenic, both individually and in a cocktail effect [15,34,92,93]. Further studies are needed to distinguish which potential trace metal exposure may connect lower IQ in children with tap water use in Afghanistan. To our knowledge, no national survey has been carried out on the subject of toxic metals in children’s fluids in Afghanistan, despite many existing hazards (unregulated industrial practices, use of leaded paint, and gasoline) [6,94,95,96]. As soon as it will be feasible, interdisciplinary studies should be performed to launch long-term longitudinal evidence-based programs involving individual, somatic, mental health, social, and environmental issues.

Older age is the second factor associated with lower IQ in children. Indeed, children’s exposure to violence and trauma-related distress is known to be associated with lower cognitive performance for schoolchildren [19,97,98,99]. Adolescence is a period when brain cognitive development is particularly sensitive to stress [100,101,102]. All sorts of stressful situations are commonly encountered in LICs and most particularly in Afghanistan, i.e., exposure to domestic and militarized violence, marital conflicts, poverty, malnutrition, poor healthcare, child labor, social pressure, and endemic diseases [103,104,105,106]. Panter-Brick et al. [13] reported that 61% of interviewed Afghan schoolchildren had suffered traumatic experiences, including loss of a close relative, witnessing violence, being in a combat zone, physical violence, or displacement [107,108]. Everyday stressors (such as debts, rental house with unpredictable rent rises, domestic violence, and a lack of family bonds) and social suffering (such as marginalization due to poverty and/or displacement, social injustice, or corruption) are closely intertwined to increase the stress load for adolescents [13,103,104,106,109,110]. Although the effective schooling of children in Afghanistan requires a strong and daily family commitment, the accumulation of stress with aging is a strong and relevant hypothesis to explain lower cognitive development for Afghan children.

The effect of gender was clearly insignificant, in spite of the fact that Afghanistan is known to suffer from a highly gendered education system that should have promoted a lower IQ result for girls [111,112]. No significant difference in learning disabilities was found between genders in Afghan public schools for children aged 6 to 12 years old [113].

The constant of the multivariate model was close to the theoretical IQ median (100), suggesting that most of the IQ variability in children had been depicted.

This study had some limitations, particularly regarding the number of schools and children involved. Access to public schools was limited because of socio-political reasons. Thus, the impacts of attaining a public school versus a private one could not be investigated. Indeed, our sampling method and inclusion had to handle highly constrained conditions in school and in the children’s randomization. Schoolchildren sampling may have underestimated the impact of stunting as stunted children are less likely to enroll in schools [30,37]. Despite this risk of selection biases, the frequency of low IQ was already high. Furthermore, this may also highlight how poor and stunted children also lack education (cycle of poverty). Our questionnaire to children was limited and could not include system-wide environmental or mental health questioning. In addition, it was not possible to interview parents to corroborate or supplement children’s responses for cultural, economic, and political reasons. Other methodologies are needed to further complete and understand these observations. Our results will enable further targeted studies as soon as they will be feasible.

## 5. Conclusions

This study showed important and original results regarding schoolchildren’s IQ in Afghanistan and provided insights on environmental exposure and adolescents’ stress burden. The use of tap water and older age were the main negative predictors for IQ while socio-economic and nutrition status showed no effect on cognitive performances of our children’s cohort. These results need to be further documented with new cohorts from Jalalabad and other Afghan cities in public and private schools, for both genders. Indeed, private schools provide the students with more learning opportunities than public schools [114,115]. Higher average student scores are found in private schools from developing and emerging countries [116,117,118]. It is also established that children in public schools belong to families with low socio-economic status who cannot afford private schools [112,115,117,119,120]. Private school development is likely to worsen the population’s socio-economic gap in Afghanistan [120], as private schools supply additional courses, more technical resources, and English language [112,121]. While our study showed no gender effect on IQ, our limited cohort calls for further investigation. Indeed, in Afghanistan, girls are prone to discrimination and learning inequalities owing to early marriage commitment, twice as much child labor as boys, lower school enrolment (mostly in public schools), and social and domestic insecurity that limit school attendance [111,120,122,123,124]. Finally, we strengthen the need for cross-sectional studies of environmental exposure to toxic elements and its relationship to the mental health and cognitive status of children. These should be launched as soon as possible to improve environment quality and the health and quality of life of children.

## Figures and Tables

**Figure 1 ijerph-19-08252-f001:**
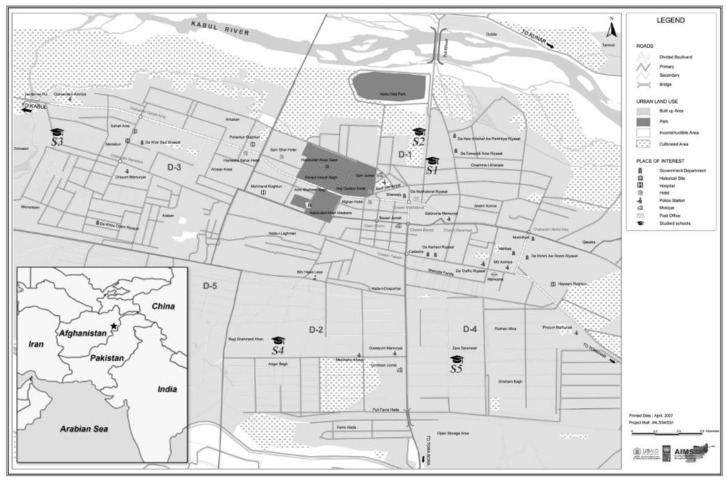
Map of Jalalabad and the studied schools with main roads.

**Figure 2 ijerph-19-08252-f002:**
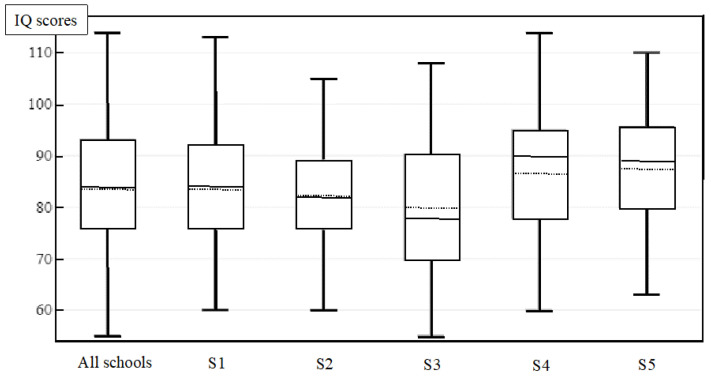
Box-whisker plots of children’s intellectual quotient for each school. Median (solid line). Mean (dot line).

**Table 1 ijerph-19-08252-t001:** Results from sample description with children’s predictors in each school. Data are means ± standard deviation, or *n* (percentage). Overweight: BMI for age z-score > 2 standard deviations.

		S1	S2	S3	S4	S5	*p*-Value
Number of students		55	41	57	50	45	
Age		11.4 ±2.0	10.8 ±2.1	12.4 ±1.9	12.1 ±2.0	11.4 ±1.9	0.001
Gender	male	38 (69.1)	41 (100)	0	50 (100)	45 (100)	<0.0005
	female	17 (30.9)	0	57 (100)	0	0	
Home near major road		32 (58)	30 (73)	35 (61)	41 (82)	28 (62)	0.06
Use of surma		18 (32)	6 (14)	20 (35)	22 (44)	10 (22)	<0.05
Tap water use		40 (72.7)	29 (70.7)	34 (59.6)	18 (36)	12 (26.6)	<0.0005
Height for age z-score		−0.52 ± 1.44	-0.30 ± 1.47	−0.28 ± 1.39	−0.76 ± 1.09	−0.46 ± 1.25	0.37
BMI for age z-score		−0.15 ± 0.77	−0.39 ±0.78	0.24 ± 1.07	0.13 ±1.10	0.10 ±1.12	<0.05
Overweight		4 (7.3)	3 (7.3)	11 (19.3)	10 (20.4)	9 (20.5)	0.12
Below poverty		19 (34.5)	13 (31.7)	23 (40.4)	23 (46)	11 (24.4)	0.23
IQ		83.7 ± 11.7	82.1 ± 10.2	79.9 ± 12.7	86.6 ± 13.7	87.2 ± 13.1	<0.05

**Table 2 ijerph-19-08252-t002:** Results from bivariate and multivariate analysis of children’s intellectual quotient with predictors. CI: Confidence interval.

	Bivariate Analyses		Multivariate Model	
Predictors	Coefficient [95% CI]	*p*-Value	Coefficient [95% CI]	*p*-Value
Intercept	Not shown		105.1 [94.1; 116.1]	<0.001
Male	−1.8 [−8.0; 4.3]	0.528	−0.9 [−6.7; 4.9]	0.729
Poverty	−3.2 [−6.5; −0.01]	0.049	−2.6 [−5.8; 0.6]	0.116
Tap water use	−3.1 [−6.3; 0.1]	0.058	−3.9 [−7.1; −0.6]	0.019
Heavy traffic road	−1.5 [−4.8; 1.8]	0.380	−1.6 [−5.0; 1.8]	0.351
Age (years)	−1.3 [−2.0; −0.5]	0.002	−1.4 [−2.2; −0.6]	0.001
Surma use	−2.0 [−5.4; 1.4]	0.252	−2.4 [−6.0; 1.0]	0.166
Overweight	−2.6 [−10.7; 5.4]	0.518	−0.2 [−8.2; 7.8]	0.956

## Data Availability

Data on Appendix A are available on request from the author. Research involving human participants and/or animals.

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
