# Peer review of "Tap Water Consumption Is Associated with Schoolchildren’s Cognitive Deficits in Afghanistan"

_ijerph, 2022, doi:10.3390/ijerph19148252_

Round 1
Reviewer 1 Report
The submission IJERPH-1764860 titled “Tap water impairs schoolchildren’s cognitive development in Afghanistan” proposes investigating the main determinants of school-aged children and adolescents of Jalalabad, Afghanistan. The manuscript is well written; even though it needs some English copyediting, it examines the factors that may influence the intellectual development of those children living in such stressful environments and with so many inequities, above all gender bias. My general comments and suggestions are presented below, and specific comments follow.
I strongly suggest changing the title to adjust to the actual results observed in the study. The way it is, it establishes causality and implies that tap water consumption causes intellectual deficits in Afghan children. In a cross-sectional study, it is only possible to raise a hypothesis, not making that statement with such rudimentary data. Thus, I suggest, “Tap water consumption is associated with schoolchildren’s cognitive deficits in Afghanistan."
I have another suggestion concerning BMI z-score as a surrogate for nutritional status. Since the authors collected data for calculating BMI, which must be described in detail in the Methods section, the use of the Height-for-Age Z-score is much more specific to chronic malnutrition; according to WHO 2009, it is an indication of stunts. Several papers successfully adjusted the association between exposure to environmental contaminants and children´s intellectual performances using this index. Here are some references in case the authors are interested:
Fonseca et al. 2008. Poor psychometric scores of children living in isolated riverine and agrarian communities and fish-methylmercury exposure. 10.1016/j.neuro.2008.07.001
WHO, 2009. AnthroPlus for personal computers Manual: Software for assessing growth of the world’s children and adolescents. WHO, Geneva
Specific Comments:
Abstract
In the population study characteristics description, please include the percentual of boys and girls.
Please change ‘exposition’ to exposure.
Material and Methods
Please check the sub-item numbering.
It is not clear how environmental exposure and socioeconomic data were collected. To whom was the questionnaire applied, and how. In the discussion session, it stated that parents were not possible to contact for several reasons. I wonder if a five-year-old child could provide reliable responses to questionnaires.
Please provide detailed information on how anthropometric measurements (height and weight) were obtained. Please include scale brands and procedures.
Results and Discussion
I would include a first table with the overall study population characteristics by the five schools with ANOVA and Post-hoc test (Tuckey) showing if the study sites (schools) were homogenous or not since it is stated that school 3 is public with a significant imbalance in the proportion of girls, besides the SES difference.
It is not clear in the multivariate modeling that the covariates ‘Overweight’ was a dichotomous variable, being overweight Yes or No. How was this variable recodified? Since only 12.7% were overweight, it would make up only 31 children. Were they evenly distributed in the five schools?
I am very concerned with the final statement “As well, it was not possible to interview parents to corroborate or supplement children's responses, for cultural, economic and political reasons and other methodologies are needed to further complete and understand these observations.” It was the children who responded to the questionnaire. The validity of the data is pretty much doubtful because of how an underage could provide information for economic status classification and other important information. And how written informed consent was obtained since the legal guardian was not adequately educated about the study?
Author Response
"Please see the attachment"

Reviewer 2 Report
Dear authors,
Thank you for your opportunity to review this manuscript.
The research topic is very important as well as precious data collected in Afghanistan.
Overall, the manuscript is well-written and sophisticated.
However, I would appreciate if the authors could add more clarifications in methods, and results.
Methods: PPI refers to Progress out of Poverty Index developed by Gramine foundation.
The paragraph of PPI is not really clear: The authors need to explain “Purchasing Power Parity” (PPP)index in order to standardize difference of currency for assessing extreme poverty (below international standard 1.25 dollar PPP). Also, the authors need to explain cut off score of PPI. PPI scores only show the probability of the international standard (for example, 25 of PPI means 40 % of likelihood of lower poverty line) if my understanding is correct.
https://tools4dev.org/resources/progress-out-of-poverty-index-tool-review/
Results:
My main concern is why the bivariate analysis between tapwater and IQ was non-significant but multi-variable analysis was significant. It may be because of external factors?
My suggestion is to add descriptive table for the all demographic variables (N, %, and group difference in IQ using Chi squire test or ANOVA) as Table 1.
Although authors mentioned Table 2, there is no Table 2 I found.
Thank you.
Author Response
"Please see the attachment"

Round 2
Reviewer 1 Report
The authors have addressed all the issues raised by this reviewer, which improved the quality of the manuscript substantially.
In the first table, the proportions of boys and girls do not sum up. Please change to 69.1 and 30.9%, respectively.
Author Response
Response to second revision (minors)
We have corrected Table 1 according to reviwer comments. This table has been inserted in the text and provided in the "fig tab second revised" file.
We have also read the manucript and corrected it for minor typos and some wrong English sentencing.
Some references were adjusted to the proper format.
This manuscript is a resubmission of an earlier submission. The following is a list of the peer review reports and author responses from that submission.